# From Positive to Idealistic: A Methodological Critique of Positive Psychology for Better Research on Idealistic Mentalities in Chinese Spiritual Traditions

**Yuchun Liu, Xintong Dong, Haohao Zhao, Jingyi Zhou and Xianglong Zeng \***

Beijing Key Laboratory of Applied Experimental Psychology, National Demonstration Center for Experimental Psychology Education (Beijing Normal University), Faculty of Psychology, Beijing Normal University, Beijing 100875, China
**\*** Correspondence: xzeng@bnu.edu.cn

**Abstract:** Chinese spiritual traditions such as Buddhism, Confucianism, and Taoism all emphasize the cultivation of idealistic mentalities (IMs) which are (1) not yet achieved, (2) clear in value judgment, (3) systematic and stable, and (4) cultivated with systematic training. While IMs are of interest to positive psychology, the methodology of positive psychology limits research on IMs. Fundamentally, positive psychology focuses on widely existing positive concepts and emphasizes being value-free, which conflicts with the features of IMs. Positive psychological studies relevant to IMs also suffer from methodological limitations: (1) recruiting samples without a spiritual background (realistic assumption); (2) ignoring qualitative differences between levels of actualization of IMs (linear assumption); (3) dividing systematic mental patterns into separate elements (reductionism); and (4) lacking value clarification during interventions. In summary, this article illustrates the methodological limitations of positive psychology in research on IMs. It encourages further research on IMs and supports the necessity of developing a new idealistic psychology for better research on IMs.

**Keywords:** spirituality; Chinese culture; Buddhism; Confucianism; Taoism; positive psychology; idealistic psychology; theoretical psychology



## 1. Introduction

Proposed by Seligman in 2000, positive psychology is 'a science of positive subjective experience, positive individual traits, and positive institutions' (Seligman and Csikszentmihalyi 2000) that has rapidly become popular around the world. It has accumulated many empirical studies and has been widely used in improving happiness, exploring well-being, and building mental conditions that enable a good quality of life (Park et al. 2016).

The relationship between positive psychology as a worldwide movement and different cultures around the world is an important topic (Delle Fave and Knoop 2013). Chinese spiritual traditions such as Buddhism, Taoism, and Confucianism have influenced several aspects of positive psychology. For example, in an early and fundamental work of positive psychology, Peterson and Seligman (2004) developed the values in action (VIA) framework to clarify universal human virtues and strengths. They investigated eight major spiritual traditions of the world, including Buddhism, Confucianism, and Taoism. Then, they proposed six virtues shared across the eight traditions, namely, courage, humanity, justice, temperance, transcendence, wisdom and knowledge, as well as 24 character strengths that underlie those virtues. Notably, the VIA framework emphasizes the virtues common to different cultures while ignoring certain virtues that are revered by a particular culture (such as revenge, which is revered in Confucianism; see Li 2015 for more; Peterson and Seligman 2004). This is consistent with the fact that positive psychology was designed as a universal science that is culture-free, which means that it should be able to 'identify and

study mental strengths and virtues that are valued by persons regardless of their culture' (Seligman and Pawelski 2003).

However, the notion of being culture-free has been criticized by some researchers who have pointed out that perspectives, concepts, and measurement tools often reflect Western culture and ignore Eastern culture. For instance, Banicki argued that positive psychology is pervaded by liberal individualism (i.e., emphasizing personal well-being; see Banicki 2014 for more perspective), which emerged from Western culture and is far from universal. As another example, one of the core concepts, subjective well-being, is defined as high life satisfaction, relatively high positive affect, and relatively low negative affect (Diener 1984). However, Joshanloo (2013) pointed out that compared to Western culture, which places more emphasis on satisfaction, Eastern culture places more emphasis on contentment and the balance between pain and pleasure. In contrast, Western culture emphasizes more positive emotions and fewer negative emotions. Thus, it is obvious that this definition of subjective well-being is Western rather than universal. Regarding measurement tools, the structure of the VIA Inventory of Strengths (VIA-IS) cannot be replicated in Indian, Chinese, and other cultures, and some items are not suitable for specific cultures (Duan et al. 2012; Ho et al. 2014). In Eastern cultures, especially Chinese spiritual traditions, the localization of positive psychological concepts is necessary. As cultural differences have been recognized, many specific concepts and exercises rooted in Chinese spiritual traditions have been incorporated into positive psychology in recent years. For example, Zeng proposed appreciative joy, a positive psychological concept rooted in Buddhism and China, as the 25th character strength (Zeng et al. 2020). Moreover, loving-kindness and compassion meditations from Buddhism have been used as important practices to increase positive emotions (Zeng et al. 2015) and other positive psychological constructs (e.g., resilience; Hansen et al. 2021). In summary, Chinese spiritual traditions provide unique ideas, concepts, and exercises for positive psychology, and they also require the localization of theory, concepts, and measurements in positive psychology, which can be considered a part of indigenous psychology (Jahoda 2016).

While previous studies have focused on the components of positive psychology (i.e., theories, concepts, exercises, and measures), the methodological issues of positive psychology related to Chinese spiritual traditions have been ignored. The current article intends to explore how the methodology of positive psychology fails to capture certain positive mentalities in Chinese spiritual traditions, that is, idealistic mentalities (IMs), which are desirable but have not yet been achieved. Notably, the methodology of positive psychology has been criticized by some scholars. For example, the empirical methods of positive psychology make it impossible to study the epistemological and axiological content of other traditional humanistic psychology concerned with positive psychology (Peng and Liu 2009). Miller (2008) argued that the foundational arguments of positive psychology involve logical fallacies such as circular reasoning, tautology, and unjustified generalization. However, no scholars have argued that the methodology of positive psychology limits research on IMs emphasized in Chinese spiritual traditions. It is worth noting that although IMs are emphasized in Chinese spiritual traditions, this study focuses more on the features of IMs rather than on Chinese culture. This differs from the perspectives of cultural psychology or indigenous psychology, which emphasize how to capture existing mentalities in certain cultures (Jahoda 2016). The sections below first illustrate the features of IMs with examples from Buddhism, Confucianism, and Taoism. Then, we argue that fundamental methodological principles in positive psychology are mismatched with these features, and we provide examples of previous positive psychological studies on related concepts. Finally, we introduce a framework of 'idealistic psychology' as a modification of positive psychology at the methodological level.

**2. Features of IMs in Chinese Spiritual Traditions**

*2.1. Not Yet Achieved*

The first feature of IMs is that they have not yet been achieved by human beings. In other words, unlike mentalities in positive psychology that are commonly experienced by many people (e.g., gratitude), the IMs discussed in Chinese spiritual traditions are considered not to exist in daily life. More rigorously, we can at least say that those IMs have not been observed among people in existing psychological research and application.

For example, Confucians cultivate the achievement of oneness, which means that one considers the universe as one's self and thus serves the world from the perspective of the universe without selfishness (Zhou et al. 2021). While ordinary people might also consider themselves part of the universe (Diebels and Leary 2019), it is obvious that it is very difficult to achieve extreme requirements such as the elimination of selfishness. Confucians proposed such oneness as the goal of training (Fung 2015), but it is not surprising that no verifiable history or empirical study has recorded the story of anyone achieving it. Similarly, Buddhists train themselves for nirvana, which is a state that includes a series of perfect qualities, such as 'unshakeable freedom of mind' (Harvey 2000, p. 41) and 'unblemished morality' (p. 40). Furthermore, Buddhists cultivate the four immeasurables, which are four prosocial attitudes (loving-kindness, compassion, appreciative joy, and equanimity) held equally toward all beings without discrimination between friends and enemies (Sears and Kraus 2009; Harvey 2000, p. 105). While Buddhists believe that nirvana and the ideal of the four immeasurables were achieved by Buddha (Harvey 2000), no empirical study has reported that any living participants have fully achieved these stages. Taoism also discusses its IM as being 'one with the Tao' (Fung 2015, p. 203) who 'transcends the distinction of things' (p. 202) as well as 'the distinction between the self and the world' (pp. 202–3). It is notable that this state is mentioned in an allegorical story called The Happy Excursion, which means that no one has claimed that anyone in history has achieved it. There are other IMs in these traditions that are not further illustrated here. Considering that these IMs are often illustrated by perfect people in fictional stories or unverifiable histories, it is fair to claim that IMs are perfect states that have not yet been achieved by human beings, or at least have not been observed in existing psychological research and practice. Furthermore, this means that these IMs are subjectively and artificially designed by spiritual traditions that are essentially different from objective mentality or behaviors that naturally exist in the real world.

While IMs have not yet been achieved, they are believed to be achievable by human beings. This could be best supported by the fact that, regardless of the origin of the idea in Buddhism, Confucianism, or Taoism, these IMs are believed to be acquirable by perfect people rather than gods or non-human beings. In Confucianism, Mencius emphasized that 'everyone can become a Yao or Shun' (two kings who Confucians believe had perfect virtues) as long as they fully develop their nature (Fung 2015, p. 466). In Buddhism, Buddha is considered an enlightened man, and Buddhists believe that some people in history have achieved nirvana, as noted above (Harvey 2000). In Taoism, even though IMs such as being 'one with the Tao' are discussed in fictional stories, it is notable that the characters in those stories are still human beings rather than gods or members of other species (Fung 2015, p. 203). This means that IMs are believed to be achievable by human beings. This idea is very important because if it is impossible to turn IMs into reality, it is meaningless to discuss them in psychology, which emphasizes empirical studies.

*2.2. Clear in Value Judgment*

The second feature of IMs is that they contain clear value judgments about what is good or desirable. Logically, since IMs have not yet been achieved, only when they are considered good or desirable are they worthy of design and further pursuit with much time and effort. Thus, clear value judgments are of vital importance for IMs.

IMs in different Chinese spiritual traditions also involve different value judgments. For example, the ultimate goal of Buddhist training is to cease suffering (Lamberton 2015),

which is similar to the hedonistic tradition in Western philosophy (Zhou et al. 2021). Notably, Buddhists try to abandon worldly happiness; thus, Buddhism is of course not secular hedonism. However, according to the category of ethics in philosophy, a philosophical school such as Buddhism is categorized as hedonism in a broad sense because of its fundamental consideration and ultimate goal of focusing on happiness and suffering (Hidalgo 2021). In contrast, Confucians emphasize morality over happiness and advocate the direct pursuit of benevolence and righteousness (Fung 2015), which is similar to the eudaimonia tradition in Western philosophy (Zhou et al. 2021). As another example, Buddhists believe that the distinction between friends and enemies is an illusion; thus, the four immeasurables in Buddhism emphasize equal prosocial attitudes toward all beings (Gold 2006). In contrast, Confucians believe that differentiated social relationships are important for human nature and society; thus, they advocate for hierarchical benevolence, such as loving one's parents more than strangers (Fung 2015; Li 2010; Moon 1996). Notably, Buddhism and Confucianism have been well integrated in many ways throughout history (Fung 2015). However, as mentioned above, they differ to some extent in terms of value judgments. Taoism and Confucianism also have different value judgments. Taoism emphasizes *Te*, which can be considered as the nature of human beings and things (Fung 2015). In Taoism, the life that follows *Te* lies beyond the distinctions of good and evil (Fung 2015). In contrast, Confucianism pursues virtues such as human-heartedness and righteousness, which represent a degeneration from *Te* in the perspective of Taoism (Fung 2015).

In summary, different spiritual traditions pursue different IMs, and different IMs contain different value judgements. Thus, universal IMs adapted to all situations and individuals do not exist, and value judgements are of vital importance for IMs. The fact that different positive mentalities have different and even contradictory values is not surprising. Schwartz and Sharpe (2006) pointed out that character strengths in positive psychology conflict with each other (e.g., white lies involve conflicts between kindness and honesty), which is not taken seriously in positive psychology. The differences in value judgements between different IMs are more obvious, so the clarification of value judgements is more important when researching IMs, which is discussed later.

### 2.3. Systematic and Stable

Another important feature of IMs is that they are systematic rather than piecemeal. As noted above, oneness in Confucianism starts with the ontological belief that the universe is the large self and human beings are part of the large self. This ontological belief further leads to value judgments, such as the idea that one should face death calmly because death is a return to the whole universe (Wang 2020; Zheng 2015). As another example, Buddhists advocate for equal loving-kindness (i.e., value judgment) because they emphasize that the difference between the self and others is illusory (i.e., ontological belief; Harvey 2000). That is, IMs in Chinese spiritual traditions are often systematic mental patterns composed of interconnected ontological beliefs and value judgments, which are more complicated than piecemeal ontological beliefs or value judgments.

In addition to being systematic, IMs emphasize stable and long-term behavioral patterns rather than short-term experiences. For example, the exemplary Confucian moral person, junzi, not only possesses all the cardinal virtues but also executes virtuous acts consistently over his or her entire life (Ip 2009). From this point of view, Confucianism not only requires understanding all the concepts but also stresses relevant acts, i.e., 'the unity of knowledge and action' (Lu 2019, p. 197). Similarly, while Buddhist-derived interventions in modern psychology can enable people to experience the non-self temporarily through meditation (Van Gordon et al. 2019), Buddhists believe that the real achievement of the non-self is everlasting; that is, one always behaves in line with the non-self in daily life rather than the non-self being a temporary experience or conceptual understanding (Fung 2015). Thus, IMs are not short-term experiences but stable and long-term behavioral patterns.

### 2.4. Cultivated with Systematic Training

Finally, IMs are often cultivated by long-term and systematic training in Chinese spiritual traditions by human beings. For example, Buddhists spend several years participating in Buddhist practice, which consists of 'right vision, conception, speech, conduct, livelihood, effort, mindfulness, and concentration', i.e., the Eightfold Noble Path, and will even give up their secular lives to do so (Kumar 2002, p. 41). Similarly, Confucianism proposes 'eight minor wires' in the Great Learning, which are the eight steps in the spiritual cultivation of the self (Fung 2015, p. 575). Taoism also has complex training to achieve absolute happiness and thus become the sage, which is illustrated in the classic book Chuang-Tzu (Chuang-Tzu and Fung 2012). Importantly, the requirement of systematic training means that these IMs cannot be obtained naturally in daily life. This implies that IMs are learned mentalities based on cultures rather than congenital mentalities based on human instinct.

Furthermore, the feature of needing systematic training entails the accumulation of repeated practice; however, it is notable that the outcomes and practices are not necessarily linear. In fact, Chinese spiritual traditions pay great attention to *Wu* (enlightenment) (Fung 2015). For instance, Buddhism emphasizes that 'for Buddhahood to be achieved, this cultivation must be climaxed by a Sudden Enlightenment' (p. 479), which is compared to 'the leaping over of a deep chasm' (p. 458). 'Either one makes the leap successfully, in which case one reaches the other side and thus achieves Buddhahood in its entirety in a flash, or one fails in one's leap, in which case one remains as one was. There are no intermediate steps between' (p. 458). The accumulation of practice is considered a sort of preparatory work (p. 479). Before and after enlightenment, there are different states, called *Jingjie* in Chinese, and nirvana, which has been mentioned before, is a state in Buddhism (p. 461). Between the different states, there are not only differences in the amount of training but also qualitative differences in the spiritual level. Thus, as mentioned later, it is improper to use the findings of partially achieved IMs to predict the findings of fully achieved IMs.

## 3. Methodology of Positive Psychology Mismatches IMs

### 3.1. Fundamental Methodological Principles

As mentalities that are positive and desirable in Chinese spiritual traditions, IMs are of interest to positive psychology. However, the features of IMs conflict with the fundamental methodological principles of positive psychology.

First, researchers have claimed that positive psychology is an objective science (Seligman and Csikszentmihalyi 2001; Friedman 2008). Positive psychology is designed to follow logical positivism (Friedman 2008; Waterman 2013) in assuming the existence of a tangible reality that can be objectively understood and measured (Park et al. 2020). Thus, the objectivity of positive psychology is emphasized, and positive psychology is distinguished from humanistic psychology, which follows post-positivism in rejecting quantitative research as a 'scientistic' distortion of science misapplied to studying humans (see Friedman 2008 for more). Consistent with logical positivism, positive psychological research prefers quantitative research with large samples (Waterman 2013). That is, positive psychology requires that the mentalities to be studied exist, or even widely exist, among the population. However, one important feature of IMs is that they have not yet been achieved. This feature logically requires that research starts not from objectively observing existing mentalities but from scholars' subjective design, followed by subjective efforts to realize that design. As Zeng (2021) suggested, empirical research on IMs follows the steps of design, actualization, and assessment. Such subjective design and creation of reality are distanced from objective science and logical positivism. Therefore, in principle, positive psychologists must wait for IMs to become existing mentalities before they can start objective and descriptive research; however, in such cases, the IMs are no longer idealistic. That is, IMs logically cannot fall within the scope of positive psychology.

Second, positive psychology is designed to be descriptive rather than prescriptive (Seligman and Csikszentmihalyi 2001). This means that positive psychology explores the results of a positive concept rather than arguing about what is positive and what is nega-

tive, which makes it a value-free science. In fact, scholars have pointed out that positive psychology is not value-free, which is not illustrated here (Banicki 2014; Prinzing 2021; Christopher and Hickinbottom 2008). In this article, we aim to show that the features of IMs make it impossible to maintain a value-free and descriptive position in studying them. As illustrated above, IMs in Chinese spiritual traditions involve clear value judgments that may differ or even contradict each other. Thus, positive psychology must provide its own prescriptive argument about why a certain IM is considered desirable rather than assuming that it has been widely approved as desirable and taking only the responsibility of description. More importantly, when studying an existing mentality, scholars can claim that they are simply describing the objective target and its impacts without judging whether the target is good or bad. However, in regard to IMs, scholars must provide a prescriptive argument for why a particular IM is worth achieving. Therefore, it is difficult to maintain a descriptive and value-free stance when studying IMs.

### 3.2. Specific Study Design or Paradigms

In addition to the fundamental methodological principles, the current section further discusses four typical methodological issues in terms of study design and paradigms that have appeared in some (although not necessarily all) positive psychological studies on mentalities rooted in Chinese spiritual traditions. Some studies have focused directly on typical IMs that do not exist for most human beings, such as oneness. Others have focused on mentalities that have not yet been acquired by the target group, such as mindfulness, which helps to illustrate the potential problems of these methodological issues in regard to IMs.

### 3.2.1. Realistic Assumption

The first methodological issue is 'realistic assumption'. As mentioned above, in Chinese spiritual traditions, it is very difficult to fully achieve an IM, even with intensive training. However, some researchers have improperly assumed that an IM has been acquired by normal people and thus have conducted empirical studies with participants who have not achieved IMs or undergone the relevant training. For example, Diebels and Leary (2019) recruited participants from the general public and measured how easy it was for them to believe that 'everything is one' (i.e., oneness). However, such a measure cannot provide a clear conclusion about believing or not believing in oneness. As mentioned before, IMs are systematic mental patterns, cultivated by systematic training. Even though some participants gave extremely high scores, whether they truly believed in oneness is questionable, as is whether they correctly understood what oneness means, considering that they had not received relevant training or education in the concept. In fact, Diebels and Leary (2019) also noted in their limitations section that their Western participants may not have thought about oneness previously or may have been only minimally familiar with this idea; thus, they suggested conducting future studies in an Eastern culture, where 'oneness is certainly more pervasive' (p. 471). However, even in an Eastern culture, such as Chinese spiritual traditions, participants without specific training or education may not currently understand the complicated idea of oneness. Therefore, Diebels and Leary (2019) provided a contributory initial exploration of oneness, but such an empirical study among participants without the necessary training or education cannot clearly illustrate 'the psychological implications of believing that everything is one', as the study title claimed, let alone the effect of achieving oneness.

Notably, not all positive psychological studies on IMs have suffered from realistic assumptions, as some recent studies have tried to cultivate IMs with interventions. For example, one study assessed a six-week mindfulness-based positive psychology intervention that briefly discussed the idea of oneness in Confucianism (Zhou et al. 2021). Another multi-week intervention of cognitive-based compassion training delivered Buddhist philosophy (i.e., discrimination between friends and enemies is illusory) and tried to cultivate equal prosocial attitudes (e.g., Ash et al. 2021). While these short interventions have tried

to cultivate IMs, none of these studies have directly evaluated whether they successfully cultivated an IM. It goes without saying that they could not evaluate the impact of a well-cultivated IM as successful cultivation of IMs in Chinese spiritual tradition is believed to be very difficult. These recent studies have nevertheless taken a proper direction in trying to cultivate IMs rather than assuming that they are already available (i.e., realistic assumption).

### 3.2.2. Linear Assumption

Closely related to realistic assumption, the second methodological issue is 'linear assumption'. As mentioned above, Chinese spiritual traditions emphasize systematic training and believe in *Wu* (enlightenment) as well as qualitative differences between different states. However, positive psychological studies often assume a linear relationship between IMs and other variables and thus use the findings of a partially achieved IM to predict the findings of a fully achieved IM, although these are actually two sides of the 'deep chasm'. For example, the abovementioned study by Diebels and Leary (2019) calculated the zero-order correlation between the extent of believing in oneness (from not believing at all to fully believing) and mental health outcomes, assuming a linear relationship between oneness and mental health. As most of the participants were not familiar with oneness, such a linear relationship further assumed that if we observe the relation between oneness and mental health among participants who do not have a strong belief in oneness or are not familiar with oneness, we can predict the relationship between oneness and mental health among participants who believe in oneness or thoroughly understand the concept. However, such an assumption ignores the sudden changes and qualitative differences in mental health among people in different states, as emphasized in Chinese spiritual traditions. Although studies on people who have achieved IMs are very rare, previous studies have implied that the training to cultivate an IM can lead to essential differences between people with and without the training, which contravenes the linear assumption. For example, meditation practitioners and non-practitioners completed the same questionnaire and had opposite results for some facets (de Bruin et al. 2012). The same meditation has a different effect on experienced meditators and novices (Nicolardi et al. 2022). Therefore, conclusions based on the linear assumptions premise may be confusing or even completely opposite.

### 3.2.3. Reductionism

The third methodological problem is reductionism, which means that an IM is simplified into elements in research. One form of reductionism is the oversimplification of a complex IM. For example, oneness in Confucianism is a combination of ontology and value judgments, as described above. However, the oneness scale in positive psychological research simply measures ontological aspects (i.e., everything is one; Garfield et al. 2014). This excessive simplification cannot represent the true connotations and state of the IM, which will affect the reliability and generalizability of the results. Another form of reductionism is evaluating independent elements and ignoring their interconnections. While there have been few empirical studies on IMs, this form of reductionism is common in positive psychological studies on other concepts. For example, mindfulness scales often measure different components of mindfulness with independent dimensions and then calculate the correlations between each dimension and mental health outcomes (Zeng et al. 2015). However, such studies sometimes lead to confusing findings. One study found that mindfulness, especially the nonjudgment of self, showed a positive relation to criminal thinking, which was explained by the failure to 'maintain fidelity to the complex concepts that underlie the construct of mindfulness' (Tangney et al. 2017, p. 1424). Another report noted that being present, when measured on a mindfulness scale, had a positive correlation with suffering, which was explained as the direct correlation between being present and outcomes but ignored the role of acceptance (Buddhists emphasize that awareness of emotions without acceptance may lead to suffering; see Zeng et al. 2015). Similarly, scholars have criticized positive psychological research for conceptualizing virtues and character

strengths as different dimensions and ignoring the relationship between them (see Banicki 2014); however, this topic is not addressed here. In summary, reductionism exists widely in positive psychology and is problematic in regard to complex concepts. As illustrated above, IMs rooted in Chinese spiritual traditions are often complex, which leads to methodological challenges in positive psychology.

### 3.2.4. Lack of Value Clarification

The fourth methodological issue in previous positive psychological studies on IMs is the lack of sufficient value clarification. As noted above, through investigating eight major spiritual traditions around the world, positive psychology identified six common virtues that were believed to be strong enough to cross cultural divides (Peterson and Seligman 2004; Seligman and Csikszentmihalyi 2001). However, positive psychology research has done little to clarify the tacit cultural and moral assumptions that in fact convey values (see Christopher and Hickinbottom 2008). Research on IMs in positive psychology also has this methodological problem. For example, many positive psychological interventions have adopted Buddhist meditations to cultivate the four immeasurables, and some have even directly pursued the four immeasurables, but they have lacked value clarification, which could lead to ethical issues (Ash et al. 2021). However, as emphasized above, IMs involve clear value judgments, which differ across spiritual traditions, indicating that IMs are based on certain cultures rather than being universal. As a result, value clarifications are needed in IM research. Only a few studies have explicitly clarified that the four immeasurables are not universal (see Zhou et al. 2021).

## 4. Better Methodology for IM Patterns

As illustrated above, positive psychology has integrated many ideas, concepts, and practices from Chinese spiritual traditions, but has not well captured the IMs that are widely discussed and intensively cultivated in Chinese spiritual traditions. Fundamentally, the principle of being objective and value-free cannot capture an IM that does not exist in the ontological sense and emphasizes value judgment. While recent positive psychological studies have tried to explore IMs with different study designs and paradigms, common methodological issues still limit research on IMs. In fact, some scholars have called for a new 'idealistic psychology' to design, achieve, and assess IMs (Zhou et al. 2021; Zeng 2021) but have not provided further arguments about its necessity or illustrated its methodological details. The current article contributes to idealistic psychology in the following ways.

First, it illustrates that IMs are widely discussed in Chinese spiritual traditions as a group of research topics that share similar features. In their empirical study, Zhou et al. (2021) simply noted that their intervention involved a non-existent mentality (i.e., oneness in Confucianism), and other studies have not highlighted IMs at all (e.g., Diebels and Leary 2019; Garfield et al. 2014). In contrast, this article points out that IMs, mentalities that are desirable in Chinese spiritual traditions but have not yet been achieved, share similar features (i.e., they have not yet been achieved, are clear in value judgment, are systematic and stable, and can be cultivated with systematic training), which supports that idealistic psychology has research topics with common features. Notably, although this article explores IMs from Chinese spiritual traditions, the four features of IMs are not limited to religions. Thus, IMs are not necessarily denominational and limited to Chinese spiritual traditions. It is possible that in the future, psychologists can also propose mentalities that are desirable but have not yet been achieved in a specific situation—that is, IMs.

Second, and more importantly, this article supports the necessity of building a new idealistic psychology from a methodological perspective. While IMs fall into the range of interest of positive psychology, as mentioned above, their features require methodological changes from positive psychology and even challenge fundamental methodological principles such as objective and value-free research. With such fundamental changes in methodology, it is reasonable to consider idealistic psychology a new kind of psychology

rather than an expansion of positive psychology. Notably, Zeng (2021) found that idealistic psychology intends to create new mentalities rather than research existing ones, which makes it fundamentally different from previous schools of modern psychology developed in Western cultures and even extends it beyond present-day psychology in terms of highlighting the possibility of psychology creating new or non-existing mentalities (i.e., idealistic mentalities). While this article does not intend to elaborate all aspects of idealistic psychology or compare it with other schools of thought, it points out that psychological research on IMs in Chinese spiritual traditions differs from the mainstream psychological research of the West (i.e., positive psychology).

Third, this article facilitates the discussion of the methodological principles of idealistic psychology. Previous discussions have emphasized only the idealistic psychology intention to design, achieve, and assess IMs, without much detail on methodological principles. This article illustrates the methodological issues in previous studies on IMs, which can help researchers in idealistic psychology to polish its methodology. For example, value clarification is important when designing the IM to be achieved, and training is necessary for participants to understand or even achieve the IM. A comprehensive discussion on idealistic psychology, such as how to achieve IMs in research, is beyond the scope of this article. Regardless of whether research on IMs is considered part of a new idealistic psychology or an expansion of positive psychology, the discussion of methodological issues will help researchers better study IMs.

**Author Contributions:** Conceptualization, X.Z.; investigation, Y.L., X.D., H.Z. and X.Z.; writing—original draft, Y.L.; writing—review and editing, Y.L., X.D., J.Z. and X.Z. All authors have read and agreed to the published version of the manuscript.

**Funding:** This research was funded by National Natural Science Foundation of China (No. 32200896).

**Conflicts of Interest:** The authors declare no conflict of interest.

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
