# Peer review of "From Positive to Idealistic: A Methodological Critique of Positive Psychology for Better Research on Idealistic Mentalities in Chinese Spiritual Traditions"

_religions, doi:10.3390/rel13111107_

Round 1

Reviewer 1 Report

This paper is very good, providing critical analysis and reflection. 

The paper discusses a large topic: psychology, beliefs, and culture, which is demanded careful and accurate analysis; however, the topic is not really new. Criticism stating that psychology is Western-oriented has been discussed a lot. The emergence of Islamic and indigenous psychology is two of many reactions toward Western-oriented psychology, including positive psychology. Please consider also mentioning this debate in the context of psychology in general. 

Please reread the paper and review the flow and style. In several parts, the paper uses too many prepositions e.g. however, moreover, etc. in sequential time. 

Examples: 

Moreover, one of the core concepts, subjective well-being, 54 is defined as high life satisfaction, relatively high positive affect, and relatively low negative affect (Diener 1984). However, Joshanloo (Joshanloo 2013) pointed out that com56 pared to Western culture, which places more emphasis on satisfaction, Eastern culture 57 places more emphasis on contentment and (especially Taoism) emphasizes the balance 58 between pain and pleasure…

Furthermore, the feature of needing systematic training means the accumulation of repeated practices; however, it is notable that the outcomes and practices are not necessarily linear. In fact, Eastern spiritual traditions pay great attention to Wu (enlightenment) (Fung 189 2015). For instance, Buddhism emphasizes that ‘for Buddhahood to be achieved, this cultivation must be climaxed

It is fine to use them as long as the flow keeps smooth. 

Please also recheck citation formatting. For example Joshanloo (Joshanloo 2013) (line 55) 

Since this paper focuses on methodology, the introduction should explore more methodological issues rather than a generic debate about Western and Eastern. The author can also show examples of measurement issues the authors considered to be problematic. 

It would be better to provide more argument on why methodological principles in positive psychology mismatch with culture? We are aware scholars/measurement creators considered cultural factors; therefore, specific points the authors view as problematic must be clearly shown. 

Be careful in developing paragraphs. A paragraph is commonly written in 5-8 sentences. Some paragraphs in this paper are too short. Double-check in this aspect is required. 

Some arguments require more evidence. For example, how did the author know the facts of the following statement:

This means that research starts not from observing objectively existing mentalities but from scholars’ subjective design, followed by subjective efforts to realize that design.

As a criticism, it is fine to state it, however, it should be supported by enough evidence. 

In summary, the idea is great. It provides criticism toward the fundamental concepts. However, as a review article, more detailed evidence is required. We understand page limits might prevent authors from providing more evidence; therefore, the selection of what the authors have to highlight should be reconsidered. 

Author Response

Thank you so much for your comments. Please find our response in the attachment.

Reviewer 2 Report

This article argues that the methodology of positive psychology limits its usefulness for research on the cultivation of idealistic mentalitics such as Buddhism, Confucianism and Taoism. There are some obvious weakness in the discourse of this article:

1.     When the authors are talking about “research on idealistic mentalities in Eastern spritural traditions, they should remember the most important principle of cultural psychology proposed by Schweder et. al. (1989): “One mind; many mentalities”. In other words, they have to consider their research problem in the context of universal mind, but not consider culture-specific mentalities first.

2.     This article argues that positive psychology “follows logical positivism in terms of epistemology” (p. 5). I think this is a wrong statement. Do not forget that Karl Popper claimed “the death of logical positivism” in 1982. If the authors follow the methodology of logical positivism, it is impossible for them to understand what means by constructing theoretical model for universal mind. For related issues on contemporary philosophy of science, please see:

Bhaskar, R.A. (1975). A Realist Theory of Science. London: Verso.

3.   For better research on idealistic mentalities in Eastern tradition, the authors should understand the cultural system approach proposed by analytical dualism. For the cultural system approach, please see:

Archer, M. S. (1995). Realist social theory: The morphogenetic approach. Cambridge, UK: Cambridge University Press.

Archer, M.S. (1996). Culture and agency: The place of culture in social theory (Revised edition). USA: Cambridge University Press.

Without a comprehensive understanding on the cultural system of Eastern tradition, it is too early to claim that “the core fracture of the IMs is that those mentalities have not yet been achieved or have not yet existed (p. 5). For an example of cultural system approach to analyze Confucianism, please see:

Hwang, K. K. (2019). Culture-inclusive Theories: An Epistemological Strategy. United Kingdom: Cambridge University Press.

4.     Because of the aforementioned weakness, the final conclusions of this article are unacceptable. For instances, the authors clam that “the current article facilitates the discussion of the methodological details of idealistic psychology.” (p. 8), but they also say that “a comprehensive discussion on idealistic psychology is beyond the scope of this article” in the same paragraph. This kind of statement make reader doubt it they really understand the “methodological details of idealistic psychology” or not.

Author Response

(The authors gave the same response as above.)

Reviewer 3 Report

Overall evaluation: The study comes out from the spiritual characteristics of Buddhism, Taoism, and Confucianism, and tries to explore the question of whether positive psychology methodology can help cultivate an idealistic mindset while focusing on the localization of positive psychology methodology in the hope of better integrating it with Eastern cultures. The authors are to be commended for the amount of reading and research they have done, however, while the reviewer understands the original intent and innovation of this study, but still feels that the paper does not do a good job of addressing why denominational spirituality must be able to be dimensionalized as an important measure of positive human psychology, and how the conflicting spiritual characteristics of the current denominations can be considered to exist in individuals and be systematically cultivated. The following queries and observations are provided for consideration.

Research flaws:

1. The abstract writing is illogical and confusing, and it does not write the research questions and innovations of this study.

2. In the section "2.1 Not yet achieved", the author summarized that Confucianism "man and the universe are one, eliminate selfishness, everyone can become a wise king", Buddhism "Nirvana ", and Taoism "unity with the Tao", but there are problems with the logical derivation of the latter part.

(1) In these religious stories, it is human beings who do everything, not gods, so in the real world, these results can happen to human beings? How is this logic reasonably deduced?

(2) The characters in religion are perfect, so such a state is usually considered unattainable, but it is also possible because it is not observed by people. Is it possible that there is no need and necessity for realization? Or is it possible that the observation and measurement of these qualities is a dimension of nothingness that prevents the realization of the self-positive and pro-social behavior that comes from the traits? Can such a concept of spirituality be localized into the conceptual paradigm of positive psychology? This is doubtful and I cannot agree with the authors.

3. In section "2.2", the author concludes that Buddhism is a hedonism that stops suffering, while Confucianism believes that morality is higher than happiness and that benevolence is the ultimate goal, but Confucianism criticizes both the immorality of Buddhist hedonism and the undifferentiated distinction between the interpersonal dimension of Buddhism, which sees no boundary between friends and enemies. The above IMs dimensions all contradict and refute each other, and the author fails to argue the necessary existence of their existence and which part is worthy to be recognized and nurtured. Since this is not argued, how can one argue that all these ideas are good? By using only stories as examples?

3. In the same section, immediately after, the author suddenly argues that these spiritual traits are related to psychological value judgments. What is the connection?

4. In "2.3", the literature concludes that Confucianism wants one to face death peacefully, while Buddhism feels that the distinction between self and other is nothing and that meditation seems to be the solution. Later, in this section, the author argues that the deeper teachings of all three sects are systematic and behaviorally relevant and that all have a systematic training method.

5. In section "3.2.1", how is unity described as a universal concept of the three religions? And, how can it be cultivated and intervened? And how can it work on the self and society? These cannot be mere ideas but need more convincing explanations.

Author Response

(The authors gave the same response as above.)

Reviewer 4 Report

The present article makes a methodological review of the current problems in positive psychology from the perspective of three representative Eastern religions. First of all, it emphasizes the existence of the Eastern religion IM and its related characteristics. Then, it comments on the descriptive and objective aspects of positive psychology, but without the spiritual background. Finally, it proposes to establish an initiative to pay more attention to idealistic psychology, which is different from positive psychology. The article has a clear structure, fluent writing and clear points of view. Give some suggestions to discuss with the author:

              1. The definition of IM is not clear enough. In Confucianism, Buddhism and Taoism, whether there is a clear definition? I don't know whether the author tends to give a unified definition, or just list three specific definitions each?

              2. On the point of "2.1 Not yet achieved"

              In my impression, is it possible that some people from the three families, such as Yan Hui of Confucianism, Lao Tzu and the Taoist priests of Taoism, as well as many Buddhas, Bodhisattvas and eminent monks of the Buddhism family, have reached such a state, but have not been seen in literature, classics or  the public? It cannot be said that "IMs do not exist "(line 134).

              Moreover, the ideal psychological state of the three families is not imaginary, it should be a state that can be realized and reached. Just not in general, or very common in the ordinary world. Or just because it is difficult to measure. Whether the reason is that Western languages are not well understood, not defined and described in appropriate language, and  so rarely known. This paper also described in line 119: While IMs have not yet had been achieved, they are believed to be achievable by human beings.

              So, is the subtitle Not yet achieved a little more specific and achieved? Not yet achieved by modern sciences? Or tended to? And so on.

              3. On "2.2 clear in value judgment"

              Line 141 &142: "Throughout history, Confucians have harshly criticized the 141 hedonistic value judgments of Buddhism as unethical".

              The goal of Buddhist practice is to cease suffering, which I think is not hedonistic, or only hedonistic, which is of great positive and social value, and is not antagonistic to Confucian.

              Buddhist is not hedonism, but because the consciousness of life reality and establish detachment, beyond earth observation, super vision, however, we all enjoy change, and all the ebb and flow of any changes in the world. Buddhist moral is transcendent to concrete and specific political life, never directly cut, interventional, buddhist only to "avidya"(无明) and "Ming(明)" to witness, his specific questions of world, Including human relations. True Buddhism does not reject morality, does not belittle Confucius and Confucianism, does not deny the world, is through the world out of the world, but in reality is often considered to avoid, escape and irresponsible, is divorced from the world, is amoral.

              The Buddhist concept of the four immeasurable minds that all beings are equal does not contradict the Confucian concept of loving relatives more than passers-by. Buddhists also pay attention to the special filial piety to parents, especially to the parents of the kindness.

              4. The content of this paper is mainly concerned of Confucianism and Buddhism, while Taoism is rarely involved. The "Oriental spiritual tradition" mentioned in the title can remind people of Hinduism and Islam, etc., perhaps only the two will be more specific?

Author Response

(The authors gave the same response as above.)

Round 2

Reviewer 1 Report

Dear Authors, 
You have done great editing, with significant changes and descriptions in the required parts. You improved a lot in referencing rules. I saw a lot of good changes in the manuscript. 
However, I will still suggest some additional information for your manuscript.
1. In the Abstract, you should also include a summary of results and what you promote. The current format is like a partial description, which is not enough to give a better picture of the research to readers. 
2. Make sure no spelling/grammatical errors. For example, related should be related to. It would be better not to put "and" to start a sentence. 

Reviewer 2 Report

In responding to the reviewers’ comments and suggestions, the author has substantially revised his manuscript and focused his discourse on the methodological limitations of positive psychology, as well as the common feature of Chinese spiritual traditions. Now I can agree with his arguments and would say that it is acceptable for publication in your journal.

Reviewer 3 Report

well done
